# BiG-SCAPE 2.0 and BiG-SLiCE 2.0: scalable, accurate and interactive sequence clustering of metabolic gene clusters

Arjan Draisma[1,3], Catarina Loureiro[1,3], Nico L. L. Louwen[1,3], Satria A. Kautsar[2], Jorge C. Navarro-Muñoz [1], Drew T. Doering[2], Nigel J. Mouncey [2] & Marnix H. Medema [1] ✉

Microbial metabolic gene clusters encode the biosynthesis or catabolism of metabolites that facilitate ecological specialization, mediate microbiome interactions and constitute a major source of medicines and crop protection agents. Here, we present BiG-SCAPE and BiG-SLiCE 2.0, next-generation methods that facilitate scalable, accurate and interactive gene cluster analyses. BiG-SCAPE 2.0 updates its classification, alignment methods, and visualizations, enabling more accurate analysis, up to 8x faster runtimes and halved memory requirements. BiG-SLiCE 2.0 updates its distance metric, pHMM database, and classification logic, resulting in increased sensitivity nearing that of BiG-SCAPE. Analysis of 260,630 biosynthetic gene clusters from publicly available genomes reveals that both tools generate concurring estimates of gene cluster diversity, thus providing significantly extended methodological support for recent evidence indicating that the vast majority of natural product diversity remains unexplored. Together, these updates will facilitate global genome mining efforts for natural product discovery and microbiome analyses scalable with current data sizes.

Microbial specialized metabolites, which showcase highly diverse chemical structures, are key mediators of ecological interactions between microorganisms as well as with their symbiotic hosts[1]. These natural products and their derivatives have been continuously exploited by mankind for, among others, their pharmaceutical and agrochemical uses[2,3]. In addition to the biosynthetic capabilities of microbes that facilitate niche adaptation through distinct biological activities, specialized catabolic pathways also allow bacteria to occupy specific niches by the consumption or conversion of specific molecules.

The leveraging of genomics data for the discovery of metabolites, enzymes and pathways, i.e., genome mining, has not only revealed a largely untapped and greatly diverse source of biosynthetic potential within genomes[4], but has also become established as a key technology to discover and exploit natural product (NP) diversity[5,6]. Genome mining for NPs, and its success, is enabled by the fact that the bacterial and fungal genes that encode the biosynthesis of specialized metabolites are frequently co-located in the genome. The signatures contained in these biosynthetic gene clusters (BGCs) can then be detected in a genomic sequence, and analysis of the biosynthetic logic encoded in a BGC can provide clues as to the chemical structure of the produced molecule(s)[7]. Computational tools, such as antiSMASH[8], have been developed to automate the identification of BGCs in genomic data, as well as the prediction of their produced chemical molecules, and have been central to the success of genome mining. Similarly, catabolic gene clusters (CGCs) that enable niche specialization and adaptation to specific dietary components or root exudates in human and plant/animal microbiomes, respectively, can be computationally

[1]Bioinformatics Group, Wageningen University & Research, Droevendaalsesteeg 1, 6708PB Wageningen, The Netherlands. [2]DOE Joint Genome Institute, Lawrence Berkeley National Laboratory, Berkeley, CA, USA. [3]These authors contributed equally: Arjan Draisma, Catarina Loureiro, Nico L. L. Louwen. ✉e-mail: marnix.medema@wur.nl

detected in microbial genomes[9,10]. Other types of gene clusters, such as those encoding exopolysaccharide production related to biofilm formation, or those encoding polysaccharide utilization pathways for degradation of complex carbohydrates, can potentially also be subject to similar genome mining strategies[11,12].

A series of studies published over the last decade have shown that clustering BGCs into gene cluster families (GCFs), based on similarities in the sets of homologous core genes, can yield useful insights into NP chemical diversity as well as the evolutionary patterns that drive BGC diversification[13–16]. In parallel, rapid developments in high-throughput sequencing have immensely expanded genome mining efforts from a few genomes to hundreds of thousands of metagenome-assembled genomes (MAGs)[4,17]. This recognition of the value and need of conducting large-scale, comprehensive investigations of the biosynthetic potential encoded in broad sets of (meta)genomes fueled the development of the Biosynthetic Gene Similarity Clustering and Prospecting Engine (BiG-SCAPE)[18] and the related Biosynthetic Genes Super-Linear Clustering Engine (BiG-SLiCE)[19].

Since 2019, BiG-SCAPE has enabled accurate and interactive sequence similarity network analysis of BGCs (and other gene clusters) and GCFs[20–22]. To do this, BiG-SCAPE combines compression of biosynthetic and other gene clusters into sets of Pfam domains[23], with a set of three distance metrics that consider domain content similarity, synteny, and sequence identity to measure (dis)similarity of gene clusters in a pairwise manner[18]. In 2021, with the inception of the extremely scalable tool BiG-SLiCE, truly global GCF analyses that include all available microbial genomes became possible[4,17,24,25]. BiG-SLiCE uses a curated library of biosynthetic profile hidden Markov models (pHMMs) to project BGCs into Euclidean space vectors, which consequently enables the use of a partitional, super-linear, clustering algorithm[19].

BiG-SCAPE and BiG-SLiCE have since established themselves as the key approaches for performing similarity and diversity analysis of gene clusters (and specifically of BGCs in the case of BiG-SLiCE), and feature as part of interoperable tools and pipelines[26–28]. Finally, this computational ecosystem that aims to identify BGCs, analyze their diversity, and predict their products' chemical structures has been further strengthened by leveraging the information contained in the MIBiG (Minimum Information about a Biosynthetic Gene cluster) repository, a manually curated set of identified BGCs with associated biochemical reference data[29]. Nevertheless, genome mining efforts continue to expand rapidly, featuring not only exponential growth of input datasets[25,30,31], but also continuous development of the concepts and data standards that provide the framework for the investigations of biosynthetic diversity and natural product discovery[8,29,32–34]. To ensure that these comparison tools continue to enable accurate and relevant analysis of biosynthetic and metabolic gene cluster data, scalability as well as biological feature driven updates are necessary.

To address these requirements, we present major new updates to BiG-SCAPE and BiG-SLiCE. BiG-SCAPE 2.0 adopts the antiSMASH (and other SMASH tools such as gutSMASH[9] and rhizoSMASH[10]) classification and 'region' concept, and adds novel gene cluster alignment algorithms that increase distance metric accuracy. In addition, it features a redesigned HTML output visualization that integrates these new biological concepts. BiG-SCAPE 2.0's Python codebase was completely rewritten to increase performance, scalability, and software sustainability; enabling analysis of dataset sizes that were previously too resource intensive, e.g., antiSMASH database[30]. BiG-SLiCE 2.0 features an updated distance metric and consequent improvement of the clustering accuracy, updates to its curated pHMM databases, added full support for antiSMASH v8[32], improved speed, and the ability to export its results directly into a tab-separated file. Benchmarking performance and accuracy improvements was made possible by a community-generated dataset of manually curated GCF assignments,

which is further made available as a resource for the wider NP research community, alongside a BiG-SCAPE 2.0 pre-processed antiSMASH DB.

These updates power accurate and efficient analysis of ever-growing datasets of gene clusters, with shorter runtimes and reduced computational resources that facilitate access to a wider range of researchers.

## Results and discussion

### New BiG-SCAPE workflows expand accessibility and scope of comparative gene cluster analysis

In light of ever-growing (meta)genomic input dataset sizes, as well as evolving concepts and data standards that guide metabolic gene cluster research, the desire to ensure that BiG-SCAPE continues to enable accurate and relevant analysis motivated substantial improvements. As such, during the development of BiG-SCAPE 2.0, we focused on improving the accuracy of the gene cluster comparisons performed by BiG-SCAPE as well as increasing performance, accessibility and ease of use. As a result, BiG-SCAPE 2.0 features four available workflows: BiG-SCAPE Cluster, BiG-SCAPE Query, BiG-SCAPE Dereplicate, and BiG-SCAPE Benchmark (Fig. 1). BiG-SCAPE Cluster comprises the main use case for the tool, i.e., the clustering of gene cluster records into GCFs, and is the equivalent of running BiG-SCAPE 1, with the exception that BiG-SCAPE 1's GCCs (gene cluster clans) are no longer computed[18]. BiG-SCAPE Dereplicate can be used to reduce computing time and redundant calculations on a dataset by carrying out a protein-sequence-based redundancy analysis to remove near-identical sequences. To do this, BiG-SCAPE Dereplicate follows the methodology implemented by the sister tool BiG-MAP[28] and leverages the fast and low-memory branchwater plugin for the FracMinHash-based tool sourmash[35,36]. BiG-SCAPE Query, formerly an optional parameter in BiG-SCAPE 1, has been expanded and lifted into its own workflow. It now allows users to provide a query gene cluster record and obtain all gene cluster records (provided by the user and/or in the MIBiG repository) that show similarity to the query, while leveraging BiG-SCAPE 2.0's complete gene cluster comparison framework. BiG-SCAPE Query facilitates rapid and scalable queries of single gene cluster records against large input datasets, in an easy-to-use workflow not yet available to the research community. Finally, BiG-SCAPE Benchmark performs a congruence analysis between a user-provided set of gene cluster to GCF assignments and the results of a BiG-SCAPE (versions 1 or 2) or BiG-SLiCE (versions 1 or 2) run. BiG-SCAPE Benchmark has been used here to test the accuracy of the two versions of both tools. In addition, we envision it can further aid users in finding the best run parameters for a given analysis (when manually curated GCF assignments are available) as well as in comparing GCF assignments between clustering tools or between sets of run parameters.

### Adopting the antiSMASH classification and 'region' concept enables more relevant comparisons with BiG-SCAPE 2.0

BiG-SCAPE has, since its inception, successfully leveraged a number of the annotations provided by the BGC detection tool antiSMASH[8]. BiG-SCAPE 2.0 continues to do so, and adds several updates that expands the usage of antiSMASH and related SMASH tools' annotation features.

BiG-SCAPE 1 defined eight custom BiG-SCAPE classes (PKS1, PKSOther, NRPS, NRPS-PKS-hybrid, RiPP, Saccharide, Terpene, Others) that encompassed all antiSMASH class/product types[18]. BiG-SCAPE 2.0 classifies records according to the legacy BiG-SCAPE 1 classes, as well as the antiSMASH class (product type, e.g., T1PKS) or antiSMASH category (broader grouping of product types, e.g., PKS) enabled by default. This allows for automated classification that does not require manual updating when using any of the SMASH tools available, such as gutSMASH[9] or rhizoSMASH[10] (Fig. 1), or when new versions of antiSMASH are released. In addition, legacy functionality is retained for users who wish to make use of the BiG-SCAPE 1 classes and their calibrated weights for the specific distance metrics[18].

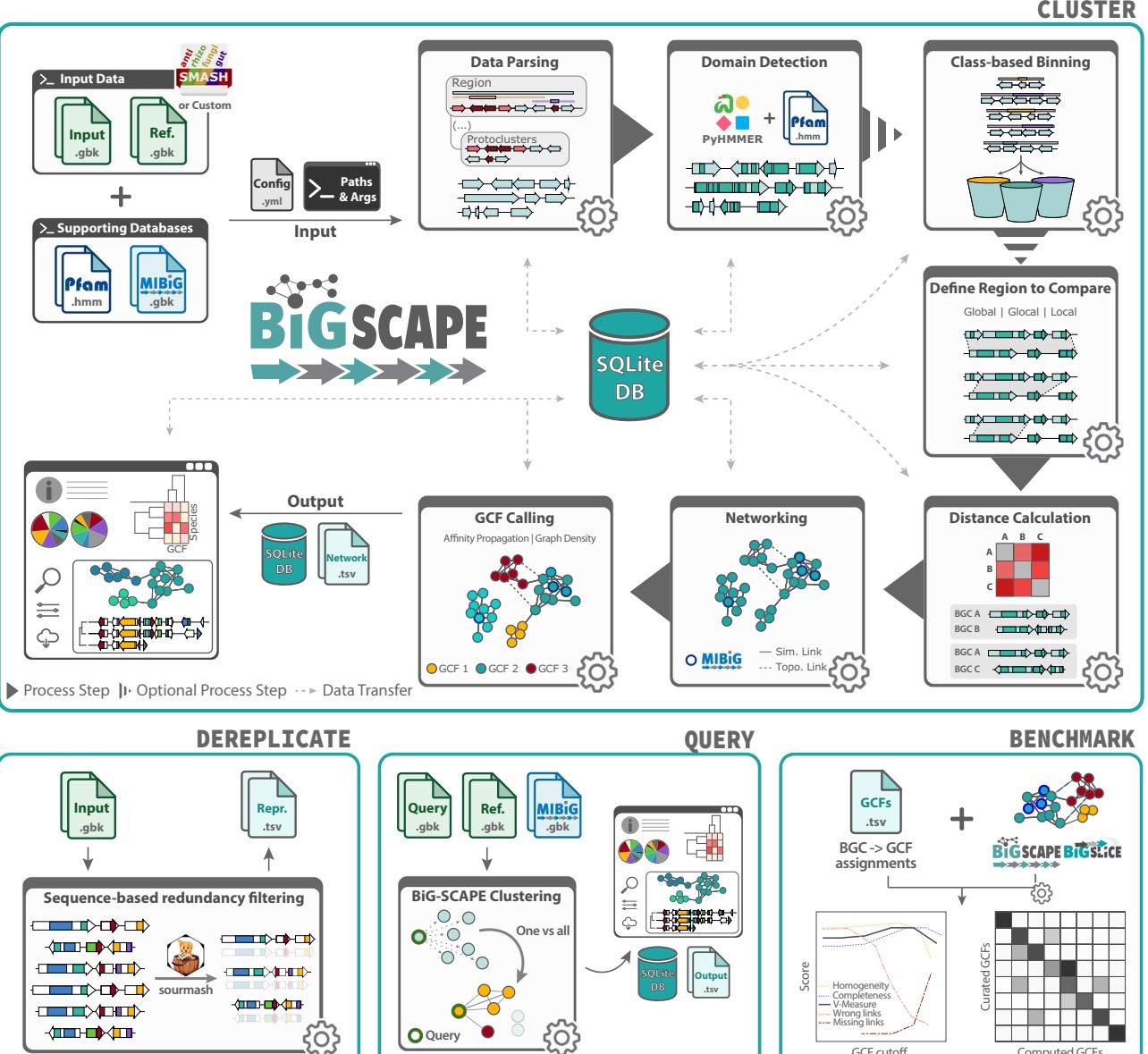

**Fig. 1 | Schematic overview of BiG-SCAPE 2.0 Workflows.** BiG-SCAPE Cluster performs pairwise comparative analysis, networking and clustering of biosynthetic gene clusters (BGCs) into gene cluster families (GCFs); BiG-SCAPE Dereplicate performs sequence based redundancy filtering; BiG-SCAPE Query performs one vs all comparisons to recover gene clusters that show similarity to a user-provided query BCG; BiG-SCAPE Benchmark performs a congruence analysis between provided gene clusters to GCF assignments and clustering results generated by BiG-SCAPE or BiG-SLiCE. The Pfam[23] logo is reproduced from the Pfam website at http://pfam.xfam.org/, licensed under CC01.0. The MIBiG[29] logo is reproduced from the MIBiG website at https://mibig.secondarymetabolites.org/, licensed under CC-BY4.0. The PyHMMER[51] logo is reproduced from the PyHMMer Git repository at https://git.embl.org/larralde/pyhmmer/, licensed under CC-BY4.0. The sourmash[35,36] logo is reproduced from the sourmash Git repository at https://github.com/sourmash-bio/sourmash, licensed under CC-BY-SA4.0. No changes were made to these logos.

Since its version 5, antiSMASH and related SMASH tools redefined the 'cluster' label and architecture to better handle co-located, hybrid, and independent gene cluster records, by introducing the *region* concept which comprises a series of hierarchically nested gene cluster record types[33]. Succinctly, the highest level of this hierarchy is the *region* record type, containing the entire sequence of an antiSMASH output GenBank file. *Region* record types may contain one or more records of the *candidate cluster* type, which in turn may contain one or more records of the *protocluster* type. A *protocluster* record contains a single record of the *protocore* type + a neighborhood at each side of the *protocore* and is defined by the antiSMASH rule that triggered detection[33]. Considering *protocluster* instead of *region* records in comparative analysis is of particular interest as it allows more specific targeting of the gene cluster records that are most likely to be responsible for the production of a given molecule, and that might have been merged into a larger region due to the greedy predictive nature of antiSMASH biosynthetic region borders/neighborhoods. This can be observed in cases such as the brasiliquinone A BGC (MIBiG BGC0002047)[37,38] from *Nocardia brasiliensis* ATCC 700358, which is detected in the genome of this strain as a T2PKS *protocluster* within antiSMASH-detected *region 7*, which contains three unrelated yet overlapping protoclusters[39]. BiG-SCAPE 2.0 has the ability to utilize any of these record types as the 'working record', while retaining the information that two or more working records (of type other than *region*) can be derived from the same *region* record by defining 'topological links'.

Finally, despite antiSMASH's success and leading role in BGC detection, which continues to motivate the leveraging of antiSMASH

and related SMASH tool annotations for gene cluster record similarity clustering analysis with BiG-SCAPE, we recognize the value in enabling input which is agnostic of the detection tool. For this purpose, BiG-SCAPE 2.0 features a '--force-gbk' run parameter, which enables any minimal GenBank file containing CDS and Sequence features to be used by BiG-SCAPE (Fig. 1). Processing such minimal GenBank files, however, will exclude all BiG-SCAPE functionality that relies on anti-SMASH annotations, such as classification, record architecture, or core domain weighted alignment (see section below) and distance calculation[18].

## BiG-SCAPE 2.0 adopts biosynthetically informed alignment strategies for handling re-arrangements and elusive gene cluster borders

Since its inception, BiG-SCAPE has been developed with strong attention to performing comparisons between the most appropriate corresponding regions between gene cluster records, i.e., defining the region to compare (Fig. 1). In practice, this means handling comparisons between complete and partial gene cluster records, or genomically adjacent gene clusters merged into one record by gene cluster detection tools. To do this, BiG-SCAPE 1 introduced a gene cluster alignment strategy, which works by finding the longest common subsequence (LCS) of pHMM domains between a pair of gene cluster records, and then using a match/mismatch penalty algorithm to extend the alignment. BiG-SCAPE 2.0 features several updates to the algorithms that perform this alignment.

Updates to the alignment modes: BiG-SCAPE 1 featured two alignment modes: 'global' and 'glocal'. Producing a 'glocal' alignment, the default option, requires three steps, finding the LCS, extending the alignment with the match/mismatch algorithm and extending the shortest arm of the gene cluster record pair on either side of the LCS to include all domains. BiG-SCAPE 2.0 adds a true 'local' alignment mode that performs the first two steps of the 'glocal' alignment (Supplementary Figs. S1, S2), and will allow comparisons of highly similar subsets of domains present in an overall divergent pair of records. This is particularly relevant in cases where the relevant biosynthetic genes are only a small fraction of the entire prediction region. Such an example can be seen with the lassopeptides citrulassins (e.g., MIBiG BGC0001550)[40,41]. Some of these, such as those from *Streptomyces davaonensis* JCM 4913 *region 5*, are detected in the context of a much larger set of genes that are irrelevant for the biosynthesis of this NP[42]. Alternatively, the 'global' mode where all domains of each gene cluster record are compared may be preferred for datasets of gene clusters with no contig breaks, and manually curated cluster borders. As the 'glocal' mode finds a compromise between the 'local' and 'global' modes, it is used by BiG-SCAPE 2 as the default mode.

Matching on protein domains, instead of CDSs: BiG-SCAPE 1 considers CDSs as the match/mismatch units. In BiG-SCAPE 2.0 the unit is now the protein domain, a change motivated by the fact that any two CDSs that share 90% of their domain content would be treated as a mismatch in BiG-SCAPE 1.

Core gene informed LCS finding: In BiG-SCAPE 2.0, the LCS finding gives preference to a core LCS, i.e., an LCS that contains core protein domains (Supplementary Fig. S2). A protein domain is considered 'core' when it is present in a core biosynthetic/catabolic coding sequence (CDS), as defined by SMASH tools as those CDSs which are responsible for gene cluster detection[8–10]. This update will minimize cases in which the 'comparable region' is centered around sets of domains that are not relevant to the core biosynthetic/catabolic regions of a record pair, or even relevant to the biosynthetic pathway at all. Transporter gene cassettes that are present in the predicted region but not involved in the NP biosynthesis are a common cause of this issue, such as those present in the region containing the paeni-nodin BGC (MIBiG BGC0001356) detected in the genome of *Paeniba-cillus sp.* FSL H7-0357 *region 2*[43,44].

Updates to the extension strategies: If an LCS has been found, the extension step may take place. BiG-SCAPE 2.0 features three match/mismatch penalty algorithm extension strategies: Legacy Extend (selected by default), Simple Match Extend and Greedy Extend (Supplementary Fig. S3). Legacy Extend follows the same principle used in BiG-SCAPE 1 and is the strictest of the strategies, with Simple Match and Greedy Extend featuring progressively simpler and less computationally intensive strategies with higher tolerance for diverse domain content and rearrangements. The impact of the differences between the behavior of these strategies using four MIBiG BGC pairs is illustrated in Supplementary Fig. S4, where the effect of translocation is particularly accentuated (Supplementary Fig. S4 and Supplementary Table S1).

## Refining GCF assignments with subgraph properties for improved clustering accuracy with BiG-SCAPE 2.0

BiG-SCAPE assigns gene cluster records to GCFs by firstly applying a cutoff to the calculated pairwise distances, which generates a gene cluster similarity network, and secondly performing affinity propagation (AP) clustering[45] to each of the connected components (CC) contained within this network. However, we, and others, have noted that AP can be overzealous in the splitting of CCs into multiple GCFs[26,46]. To address this, BiG-SCAPE 2.0 analyses each CC with regard to its degree of density, i.e., the ratio between the number of edges and the maximum number of possible edges. For any CC that displays graph density ≥ 0.85 and is thus tightly connected, BiG-SCAPE will penalize the generation of new GCFs, thus assigning its gene cluster records to a smaller number of GCFs. In practice, this is achieved by altering AP's internal preference parameter to − 5[45]. Both graph density and AP preference parameters are configurable in BiG-SCAPE 2.0's config.yml file.

## An updated BiG-SCAPE 2.0 user interface facilitates comprehensive and interactive analysis

Another of BiG-SCAPE's goals is to provide an interactive user interface (UI) that facilitates effective navigation and interpretation of the output data. In order to tackle scalability issues derived from rendering ever-growing input dataset sizes, as well as to facilitate interaction with the new features described above, the BiG-SCAPE 2.0 interactive UI underwent significant updates (Fig. 1, Supplementary Figs. S5, S6).

While the overall structure and information displayed in the UI retain the well-established look-and-feel, the Network section has been significantly restructured. This section is now organized on the basis of a connected component (CC) table. The user is able to select any CC from this table, and its subnetwork will be loaded in the well-known BiG-SCAPE interactive UI style. Interacting with the loaded (sub)networks remains consistent with BiG-SCAPE 1, as do the gene cluster detail and GCF tree panels. In addition, improved in BiG-SCAPE 2.0's UI are the filtering and selecting capabilities, where filtering elements can be compounded with logical operators, e.g., "[pHMM/Pfam domain name] AND [GCF identifier]". The filtered selections (at either CC or network level) can additionally be easily downloaded as tab-separated value (TSV) files (Supplementary Fig. S5).

Furthermore, BiG-SCAPE 2.0's interactive UI has also been updated to enable users to fully leverage the functionality offered by the newly featured antiSMASH *region* concept. For this purpose, *topological* links are introduced, which are represented by dashed line edges indicating that two gene cluster records were derived from the same antiSMASH *region* (Fig. 1, Supplementary Figs. S5, S6). Finally, as a quality of life token, BiG-SCAPE 2.0's interactive UI can now be used in Dark Mode.

## BiG-SCAPE 2.0 is fully refactored for software sustainability and interoperability

BiG-SCAPE 1 was rapidly adopted by the NP genome mining research community as one of the most widely used tools for performing

comparative analysis of BGCs. Nevertheless, as often seen in research software developed in an academic context[47–49], BiG-SCAPE 1's codebase remained of a prototypical nature, and suffered from shortcomings with regard to performance and maintainability. This motivated the complete rewrite of BiG-SCAPE's Python codebase, with a focus on code readability and modularity, implementation of CI/CD (continuous integration/continuous development) and a unit/integration testing framework (Supplementary Table S2). Considerations towards improved performance and scalability included the replacement of the stand-alone HMMER suite[50] with the Python-based HMM scanning library, PyHMMER[51], and the use of a central relational SQL database schema (Supplementary Fig. S7) implemented as a file-based SQLite database[52]. This database stores processed input as well as computed similarity and clustering results (Fig. 1), supplies BiG-SCAPE 2.0's interactive UI with all required information, and enables memory-conservative native modules such as network handling and multithreaded distance calculation. BiG-SCAPE 2.0 also supports the reuse of a single database throughout several runs with different run parameters and/or input datasets. While the Networking and GCF Calling process steps (Fig. 1) will always be computed for each run, all the data already present in the database that is required by any of the previous process steps will be re-used. This will greatly speed up reruns of (subsets of) the same input dataset,with differing parameters, or the addition of new records to already existing datasets. Finally, providing analysis outputs in an SQLite database facilitates efficient analysis requiring complex queries, can serve as a gene cluster collection information management system with scope beyond GCF clustering and enables more efficient integration with interoperable tools and pipelines[26,27].

Furthermore, with regard to software sustainability, synchronizing BiG-SCAPE 2.0 with future updates of the commonly used gene cluster detection tools has been made easier by flexibly parsing input data, e.g., by moving away from the previous hard-coded antiSMASH classification and facilitating the use of non-SMASH-tools-processed gene clusters. We have additionally generated comprehensive in-depth documentation and run tutorials that can be found on the wiki attached to the BiG-SCAPE Github page. Finally, while BiG-SCAPE is internally prioritized for long-term maintenance, continued development of BiG-SCAPE further benefits greatly from community contributions as well as user feedback, which is welcomed and can be done easily through the GitHub page.

## BiG-SLiCE New features and updates

To perform its scalable GCF calculation, BiG-SLiCE 1 transformed BGCs into numerical vectors of biosynthetic domain counts and then greedily grouped these vectors into GCFs with the BIRCH clustering algorithm[53] using the Euclidean distance metric. Since the vector calculation depends on the presence or absence of specific biosynthetic domains, certain BGC classes, such as RiPPs, produce much sparser vectors compared to others (like PKSes and NRPSes). While this disparity was mitigated in BiG-SCAPE through class-specific distance weighting, it results in significant clustering sensitivity imbalances in BiG-SLiCE, identified as one of the tool's biggest drawbacks in the original paper. In BiG-SLiCE 2, we addressed this issue by replacing the Euclidean metric with a Cosine-like distance metric, achieved by applying L2 normalization to the matrix. This adjustment ensures much more balanced clustering across all BGC classes[4,17]. In addition, the original "biosynthetic PFAM" database and BGC class definitions, previously based on PFAM v32 and antiSMASH v5, have been updated to align with the latest versions of PFAM v35 and antiSMASH v8.

Furthermore, the new BiG-SLiCE delivers a notable speed improvement, reducing the total clustering time by 25-50% (Supplementary Table S3). This enhancement was achieved through overall code optimization and the elimination of certain I/O-induced bottlenecks. Like BiG-SCAPE, the stand-alone HMMER suite has been replaced with the Python-based HMM scanning library, PyHMMER[51]. This change not only further improved speed but also simplified the installation process via pip. In addition, in response to high user demand, we have introduced a new program parameter ('--export-tsv') that allows users to export all parsed BGC metadata, vectorized features, and clustering results as tab-separated text files (TSVs). Overall, these updates align BiG-SLiCE more closely with its primary purpose: efficiently parsing large sets of raw BGC files and generating GCF assignments for integration into broader analytical workflows. Last but not least, several issues present in the original version have now been fixed, thanks to significant contributions from members of the community.

## Using curated BGC-GCF assignments to demonstrate accuracy strides made by both tools, with BiG-SLiCE showcasing largest gains

To demonstrate the impact of the aforementioned updates to the algorithms driving the biological accuracy of both BiG-SCAPE 2.0 and BiG-SLiCE 2.0, we leveraged BiG-SCAPE 2.0's Benchmark workflow, as well as 9 datasets of manually curated GCF assignments (Supplementary File 1) obtained with the aid of the wider research community. In order to assess the congruence between the curated and computed assignments, BiG-SCAPE Benchmark utilizes the V-measure[54] an evaluation metric that combines homogeneity, (i.e., whether computed GCF members are assigned to the same curated GCF) and completeness, (i.e., whether curated GCF members are assigned to one computed GCF).

For each curated dataset, we ran BiG-SCAPE and BiG-SLiCE with increasing GCF cutoffs, and for BiG-SCAPE in particular with varying alignment modes and extension strategies (see methods, Supplementary Fig. S8). As V-measure is not adjusted for chance, comparing clusterings of different sizes (i.e., with different numbers of GCFs between computed and curated assignments) can be misleading[54]. Therefore, when comparing clusterings of different versions of BiG-SCAPE and BiG-SLiCE, we focus only on the V-measure score for the GCF cutoff that produced, for each tool and set of run parameters, the number of GCFs closest to that of the curated assignments (Fig. 2).

We can see that, in the large majority of the datasets, BiG-SLiCE 2.0 showcases improved accuracy when compared to its previous version (average relative V-measure increase = 17%), with the largest increases seen in datasets particularly associated with shorter BGCs such as RiPPs (Dataset G: relative V-measure increase = 106%; Fig. 2). This improvement is expected as the cosine-based clustering helps alleviate the imbalance within the different BGC classes. With regards to BiG-SCAPE, we can see that there has been no loss in accuracy, with one or more modes of BiG-SCAPE 2.0 performing equally well or better than BiG-SCAPE 1.1 across all datasets. The different BiG-SCAPE 2.0 modes/run parameters available result in slight differences in accuracy across each dataset, indicating that there is no one optimal mode that performs best (or worst) for all datasets (Supplementary Fig. S8).

Dataset H (Reclassification) consists of only Glycopeptide NRPSs, which have highly similar gene composition and architecture, with curated GCF assignments that indicate grouping by subtypes (Supplementary File 1). The GCF boundaries being this specific against a backdrop of highly similar BGCs makes this a particularly challenging dataset to cluster for both tools, as we can see in Fig. 2. BiG-SCAPE 2.0's newly added alignment mode 'local' and extension strategy 'simple match' stands out as being able to best capture these detailed features (Supplementary Fig. S8). This is also one of the only two instances where BiG-SLiCE 2.0 performs slightly worse than its predecessor.

The other instance where this happens is with Dataset B (Divergent). This dataset includes singletons, as well as GCFs that contain BGCs with highly similar gene architecture that produce the same or highly similar metabolites, but are encoded in the genomes of taxonomically divergent bacteria (Supplementary File 2). While BiG-

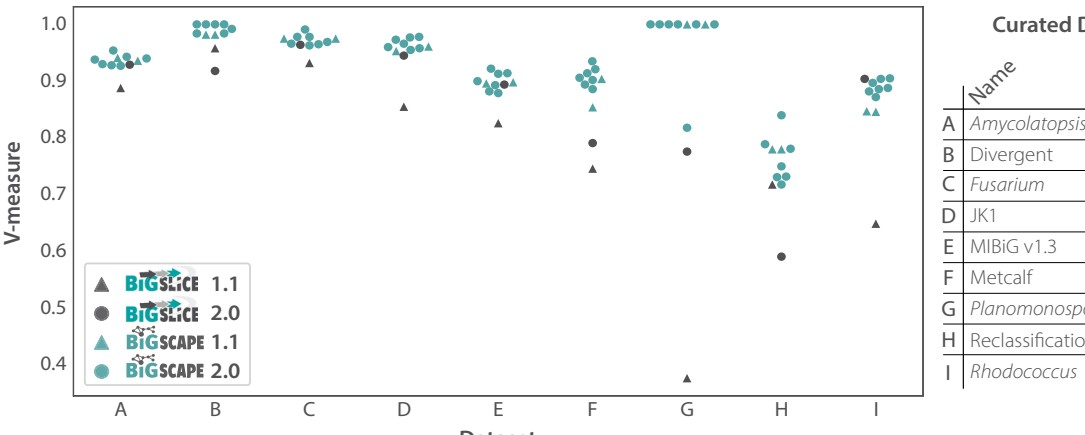

**Fig. 2 | BiG-SCAPE/SLiCE (versions 1.1 and 2.0) clustering results, measured by the level of clustering agreement (V-measure of 1.0 for perfect clustering), compared to nine datasets with curated biosynthetic gene cluster (BGC) to gene cluster family (GCF) assignments at the GCF cutoff where the number of computed GCFs best matches the number of curated GCFs.** Curated dataset name and metrics (number of BGCs, curated GCFs, and distinct antiSMASH product classes) are also shown. For BiG-SCAPE versions 1.1 and 2.0, each datapoint represents a specific set of run parameters/modes (see methods, Supplementary Fig. S8). Source data are provided as a Source Data file.

SCAPE 2.0 produces biologically relevant clusterings, BiG-SLiCE, being optimized for speed, performs slightly worse, unable to capture this diversity with as much accuracy.

Still, it is notable that BiG-SLiCE performs particularly well in dataset I (*Rhodococcus*), which has relatively large curated GCFs when compared with the other curated datasets, and features low diversity in classes (Supplementary File 1), resulting in a biosynthetic diversity context in which BiG-SLiCE performs with comparable accuracy to BiG-SCAPE.

Finally, Dataset F (Metcalf) features curated GCF assignments based on the molecular similarity of experimentally characterized metabolites produced by their BGCs (Supplementary File 2), and showcases how BiG-SCAPE 2.0 is consistently able to connect the genomic biosynthetic landscape to the chemical space of produced metabolites across run parameters.

## Benchmarking performance highlights speed improvements for both tools, with BiG-SCAPE 2.0 shining in mid-sized datasets

To benchmark the performance of BiG-SCAPE 2.0 and BiG-SLiCE 2.0 (Fig. 3), and consequently demonstrate the strides made in terms of scalability, we generated increasingly large random partitions of GenBank files from antiSMASH DB v4.0b2[30], and used 16 cores on a 750 GB RAM server (see "Methods").

BiG-SCAPE 2.0 is 2-8x faster than BiG-SCAPE 1.1, depending on dataset size, and is able to process at least three times the amount of input GenBank files handled by BiG-SCAPE 1.1 on this hardware setup (Fig. 3a and Supplementary Table S3). What is more, at input sizes up to approximately 10,000 GenBank files, which we expect encompass the most common use cases for BiG-SCAPE, BiG-SCAPE 2.0 runtimes are comparable with those of BiG-SLiCE. At the input dataset sizes and computational resources used, BiG-SLiCE 2.0 is only 1.5x faster than its predecessor, but we predict these differences to become more pronounced with larger dataset sizes, as well as with more available CPUs. As expected, at larger input dataset sizes (from approximately 10,000 GenBank files) BiG-SLiCE continues to perform at orders of magnitude faster runtimes than BiG-SCAPE, due to its hyper-scalability (Fig. 3a and Supplementary Table S3). Furthermore, we tested whether using alternative record types or extension strategies would have an impact on runtime (Supplementary Table S4). We see that runtimes increase slightly when using *protocluster/protocore* records instead of region records, which is expectable given the increase in the number of total records being considered in *protocluster/protocore* runs, as well as

overlaps between records leading to the same genomic region being accounted for in more than one record. Running BiG-SCAPE 2.0 with alternative extension strategies leads to little to no variation in runtime (Supplementary Table S4).

When analyzing the proportional runtimes of BiG-SCAPE subtasks (Fig. 3b and Supplementary File 3), i.e., the runtime for each subtask in a run as a percentage of the total runtime, we can see a major decrease in the runtime required for performing the hmmscan and hmmalign tasks. This is a result of the replacement of the stand-alone HMMER suite with the Python-based HMM scanning library, PyHMMER[51]. In practice, at input dataset sizes of 25,000 gene cluster records and above, the bulk of a BiG-SCAPE 2.0 run is spent calculating distances. This underpins the importance of the aforementioned updates that aim to minimize the number pairwise distances calculated by ensuring that only relevant gene cluster record pairs are compared.

Finally, when analyzing the CPU and RAM usage breakdown of a full run of BiG-SCAPE versions 1.1 and 2.0 (Fig. 3c), using an input dataset size of 25 000 gene cluster records, we observe that BiG-SCAPE 2 is over twice as conservative with regards to RAM usage, and more efficient with regards to CPU usage, i.e., CPU's are maximally used throughout the majority of the run.

## Analysis of 260,630 biosynthetic regions from the antiSMASH database provides new evidence for uncharted natural product chemical diversity

To further demonstrate BiG-SCAPE 2.0's scalability, as well as provide the research community with a valuable analysis resource (the provided output SQLite DB can now be easily further analyzed and re-used, see Data Availability), we processed the entire antiSMASH DB 4.0 making use of the novel *protocluster* record type feature (Fig. 3d). In detail, antiSMASH DB consists of 260,630 GenBank files (region type records) and 292,050 *protocluster*-type records; and was grouped into class-based bins of which the largest (RiPP-like) contains 32,295 *protocluster* records. BiG-SCAPE was able to process this data, using 128 cores and approximately 1TB of memory, in 89.2 h (Fig. 3d). The same dataset was processed using BiG-SLiCE 2.0 in 6.1 hours, using 64 cores and approximately 175 Gb of memory. Based on the resource use, this dataset size likely approaches the maximum for BiG-SCAPE on most standard server hardware architectures, whereas BiG-SLiCE remains a viable choice for datasets scaling into millions of BGCs. This global analysis led to the generation of 20,570 CCs and 26,260 GCFs with BiG-

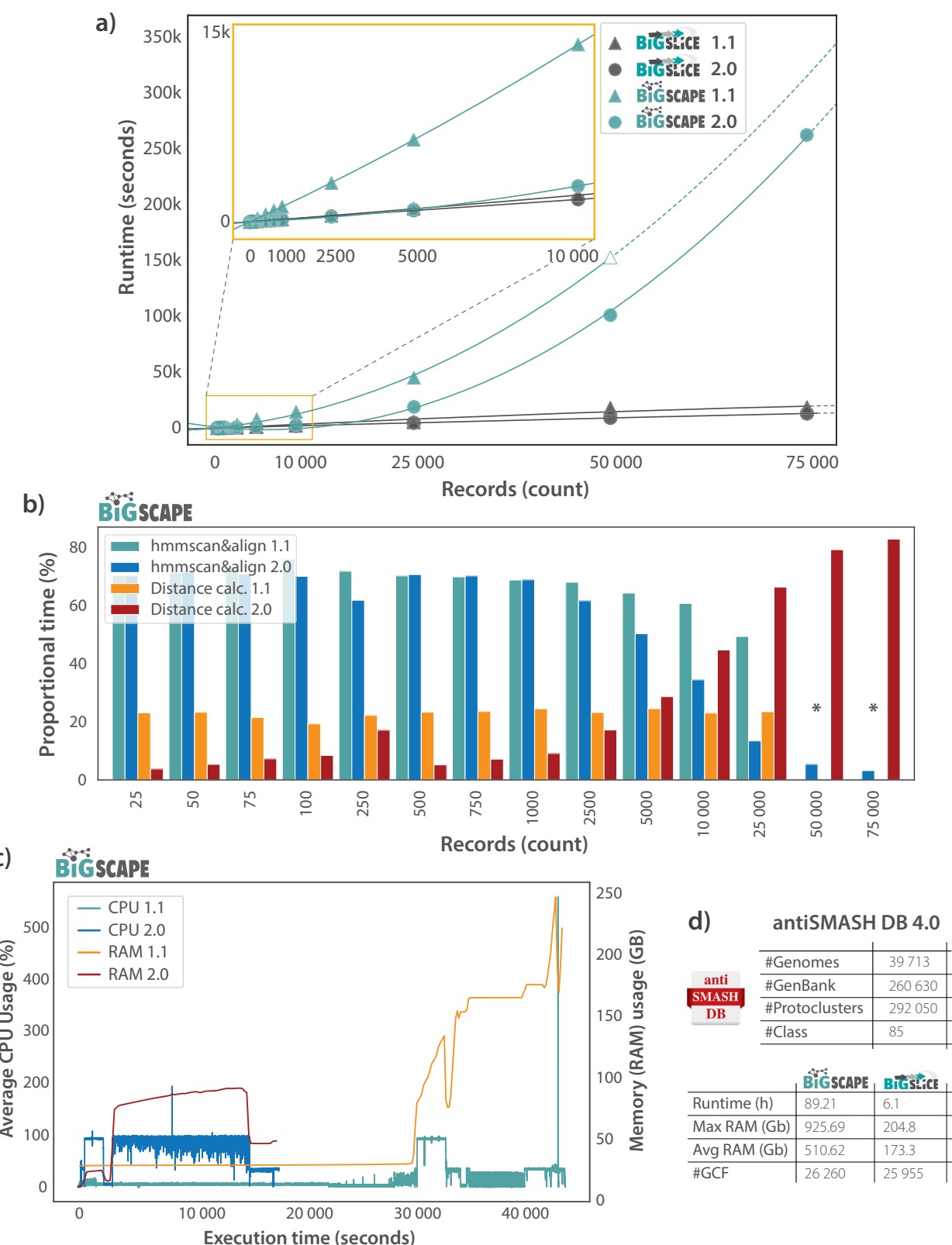

SCAPE 2.0, and 25,955 GCFs using BiG-SLiCE 2.0 (Fig. 3d). This is in agreement with previously computed maps of global diversity, i.e., in the Compendium of bacterial biosynthetic diversity, that estimate that only about 3% of bacterial biosynthetic potential has been described[4]. While we wondered whether the estimated numbers of GCFs in these global maps, based on BiG-SLiCE, might have been inflated by gene clusters being artificially merged into larger regions, or by small gene

clusters with large variable flanking regions, we can see that, in practice, this is not the case. Instead, GCF numbers estimated by BiG-SCAPE 2.0 using *protocluster* record type, are very similar to those computed by BiG-SLiCE. Hence, these results confirm that at a global scale such biases are relatively small, and that both tools, even using different record types, produce comparable estimates of biosynthetic diversity. This provides significantly extended methodological rigor to the

**Fig. 3 | Runtime and computational resources (random access memory (RAM), central processing units (CPUs)) breakdown of multiple BiG-SCAPE and BiG-SLiCE runs.** Runs in panels a-c used 16 CPUs on a 750GB RAM computing cluster. **a** Runtime comparison between multiple runs of both tools, versions 1.1 and 2.0, at increasing input dataset sizes (Supplementary Table S3). The estimated runtime of an incomplete BiG-SCAPE 1.1 run, which crashed at the final steps caused by the final spike in RAM seen in panel 3.c, is added to provide more information on how this version would scale, and is represented by an unfilled triangle (Supplementary Fig. S10, see methods). Regression lines are shown as dashed lines after the last finished run for each tool/version. **b** Proportional runtimes of two major tasks (hmmscan&align and distance calculation) performed by BiG-SCAPE versions 1.1 and 2.0, at increasing input dataset sizes. Bars indicate the runtime for each task as a percentage of the total run time (Supplementary File 3). * Indicates missing runtimes for BiG-SCAPE 1.1 for these datapoints, as these dataset sizes were too large for the tool. **c** Runtime breakdown of computational resource (CPU and RAM) usage of full BiG-SCAPE versions 1.1 and 2.0 runs with an input dataset size of 25,000 gene cluster records (Source data). **d** Run details for BiG-SCAPE 2.0 (using 128 cores) and BiG-SLiCE 2.0 (using 64 cores) runs using antiSMASH DB 4.0 as input (see methods section for further run details). Source data are provided as a Source Data file.

evidence for the vast majority of natural product chemical diversity still not having been elucidated.

Finally, while all previously described benchmarking has been carried out against earlier versions of the tools, we additionally leverage the antiSMASH database to compare our tools with the most recent alternative gene cluster clustering tool, IGUA[55], developed in parallel with BiG-SCAPE v2 and BiG-SLiCE v2. On antiSMASH-DB, IGUA emerges as the fastest (53.1 min) of the three tools with good accuracy (see Supplementary Figs. S8, S9 for details) and thus provides a useful complementary algorithm for medium-to-large datasets. It gains considerable speed by replacing the use of pHMMs by MMseqs2 clustering of protein sequences, although theory suggests that its use of an all-vs-all distance matrix and hierarchical clustering will scale less favorably with very large datasets that exceed the antiSMASH database in size. Furthermore, such choice of algorithm does not allow direct assignment of new BGCs to pre-processed GCFs, which is a very useful feature of the vector clustering approach used in BiG-SLiCE, demonstrated by the wide usage of our BiG-FAM database[24]. So altogether, IGUA provides a useful complementary algorithm with very fast clustering of medium-to-large datasets, even though BiG-SLiCE v2 and BiG-SCAPE v2 still provide unique capabilities with larger theoretical scalability and the ability to place BGCs into precomputed GCFs (for BiG-SLiCE) and interactive network visualization and exploration (for BiG-SCAPE). Future work could explore potentially merging MMseqs2-based features with vector clustering and/or sequence similarity networking to combine the advantages of each method.

### Conclusions and future perspectives

With the use of tools like BiG-SCAPE and BiG-SLiCE, genome mining for specialized metabolites can go beyond identification of novel metabolites, towards exploring the landscapes of secondary metabolic diversity at a planetary scale. Here we present BiG-SCAPE 2.0 and BiG-SLiCE 2.0, the next generation of biosynthetic genomics clustering tools.

The complete refactoring of the BiG-SCAPE codebase is an important stepping stone upon which continuous development of new features and improvement of existing ones can be built, and will additionally facilitate its integration with other tools and pipelines[26,27]. Our concern for code modularity and software sustainability has also allowed important strides to be made in terms of scalability, with BiG-SCAPE 2.0 standing side by side with BiG-SLiCE's runtimes at small and medium dataset sizes (< 25k GenBanks, Fig. 3.a).

We have further observed that, in particular with respect to BiG-SCAPE 2.0, no single optimal set of run parameters consistently leads to the most biologically robust clustering of all manually curated benchmarking datasets (Fig. 2). Therefore, to draw solid biological conclusions from this type of clustering analysis, we encourage the user to go beyond the default by investigating and adjusting the highly flexible run parameters that BiG-SCAPE 2.0 has to offer. With newly introduced features aimed at generating ever more relevant comparisons between gene cluster records, and an SQLite database that greatly reduces rerun times, BiG-SCAPE 2.0 is a tool designed to allow the user to intimately explore their data and clustering results.

Furthermore, while the concept of a Gene Cluster Family is not static across dataset characteristics and analysis objectives, we see that BiG-SCAPE 2.0 is able to match and raise the accuracy of its predecessor, and that BiG-SLiCE 2.0 makes important strides in this aspect, with accuracy levels that nearly match those of BiG-SCAPE (Fig. 2). In this way, we are confident that both BiG-SCAPE 2.0 and BiG-SLiCE 2.0 can, at their default run parameters, accurately capture intricate patterns in biosynthetic genomic space which reflect evolutionary relationships and connect it to the chemical space of their specialized metabolites.

The improvements made to both tools keep pace with developments in the natural products research field, and will continue to aid researchers in their study of microbial specialized metabolism diversity and evolution, as well as in their efforts to discover novel bioactive compounds.

## Method

### Benchmarking BiG-SCAPE and BiG-SLiCE runtimes

To construct a testing dataset for benchmarking performance, i.e., runtime and computational resource usage of BiG-SCAPE versions 1.1.9 and 2.0.0 and BiG-SLiCE versions 1.1.1 and 2.0.0, we generated increasingly large random partitions of GenBank files from antiSMASH DB v4.0b2[30] (Supplementary File 4). Partition sizes are as follows, in the number of GenBank files: 10, 25, 75, 100, 250, 500, 750, 1000, 2500, 5000, 10,000, 25,000, 50,000, 75,000. All partitions up to 50,000 GenBank files were generated in triplicate by randomly sampling. Due to a technical issue, BiG-SCAPE v1 has two partitions for samples of 5000 GenBank files, and for the partition size of 75,000 GenBank files, a single partition was generated. All benchmarks were run using 16 cores on a 750 GB RAM server with an AMD EPYC 7532 32-Core Processor @ 2,2 – 3,3 GHz, making use of a Lustre[56] network mounted file system. Benchmark runs of BiG-SLiCE versions 1.1.1 and 2.0.0 were run using default parameters. Benchmark runs of BiG-SCAPE versions 1.1.9 and 2.0.0 were run without a classification ('--no-classify' for version 1.1.9, '--classify none' for version 2.0.0) and toggling mix mode ('--mix').

Total runtimes were retrieved from output log files for BiG-SCAPE, and from SQLite database run logs for BiG-SLiCE. Runtime values for all replicate runs were averaged (Fig. 3a). Since the computational processes used in BiG-SCAPE are expected to scale non-linearly, exponential regression lines were added to plot the projected runtimes beyond the recorded runtimes using the regplot function from the seaborn Python package[57] with an 'order=2' parameter. In order to provide further insight on how BiG-SCAPE 1 could scale, if it were not to crash due to high RAM requirements, a runtime of a crashed run is represented by an unfilled triangle. The runtime of this crashed run was calculated by collecting the creation time of the last file created in the crashed run and the end times for each run under 50000 records. A time difference in seconds was calculated for these times (hereafter referred to as the "missing runtime"). Missing runtimes per number of records were used in a second-order polynomial regression fit to calculate the projected missing runtime for a run at 50000 records using the polyfit function from the NumPy Python package[58]. This projected missing runtime was added to the creation time of the last file in the

**Table 1 | Run parameter description of BiG-SCAPE 2.0.0 record type and extend strategy runtime benchmarking**

| Variable to benchmark | Run Parameters |
|---|---|
| Simple Match extend strategy | '--classify none --mix --extend-strategy simple_match --alignment-mode local' |
| Greedy extend strategy | '--classify none --mix --extend-strategy greedy --alignment-mode local' |
| Legacy extend strategy | '--classify none --mix --extend-strategy legacy --alignment-mode local' |
| Protocluster record type | '--classify none --mix --record-type protocluster' |
| Region record type | '--classify none --mix' |
| Protocore record type | '--classify none --mix --record-type proto_core' |

crashed run to arrive at a projected run end time (see Supplementary Fig. S10 and Supplementary Table S3).

Performance benchmarking of BiG-SCAPE 2.0.0 runtimes with alternative record types and extend strategies (Table 1 and Supplementary Table S4) was done using the 10000 GenBank file antiSMASH DB v4.0b2 random triplicate partitions, and all replicate runtime values were averaged.

**Benchmarking BiG-SCAPE proportional task runtimes**

Runtimes of the major tasks performed by BiG-SCAPE versions 1.1.9 and 2.0.0 (Fig. 3b and Supplementary File 3) were directly obtained from output log files for BiG-SCAPE 2.0.0, and from inspection of modification dates of output files produced during different steps of the workflow for BiG-SCAPE 1.1.9 (where runtimes or timestamps for specific subtasks are not provided in the log files) (Table 2). The major subtasks consist of input parsing, hmmscan, hmmalign, distance generation and GCF calling. Modification times on the file system used are recorded with a maximum resolution of seconds. Due to this, runs using 10 records were excluded for task runtime analysis as individual tasks could not reliably be distinguished for BiG-SCAPE 1.1.9.

Proportional runtimes of the described tasks were then calculated by dividing the absolute runtime of each task in the analysis by the logged total runtime of the full analysis. It must be noted that a fraction of the total runtime remains unaccounted for, which primarily corresponds to time spent on output generation that is not captured by the defined tasks.

**Benchmarking BiG-SCAPE computational resources**

To obtain a runtime breakdown of computational resource (CPU and RAM) usage of BiG-SCAPE versions 1.1.9 and 2.0.0, a profiler was built into BiG-SCAPE 2.0.0, and back-ported into BiG-SCAPE 1.1.9. The profiler data was further complemented with data provided by the Ganglia computer monitoring software[59] active on the machine used in the benchmarks. Resource usage of a run using partition 25000_1 (Fig. 3c and Supplementary File 4) of 25000 GenBank files by BiG-SCAPE versions 1.1.9 and 2.0.0 was analyzed to produce Fig. 3.c. Differing strategies for multiprocessing and parallelization of subprocesses between BiG-SCAPE versions 1.1.9 and 2.0.0 resulted in duplicate timepoints, and CPU usage data for any duplicate timepoints was averaged. Obtaining accurate RAM usage from the profiler was not feasible due to technical issues, thus RAM usage data was retrieved using the Ganglia software, active throughout the benchmark runs. When selecting the time span for resource monitoring, minutes were rounded down for start time and rounded up for end time.

**Analyzing the antiSMASH database 4.0**

We downloaded the full antiSMASH DB v4.0b2 in JavaScript Object Notation (JSON) format[30], and subsequently processed it with antiSMASH 7 to obtain GenBank files. This dataset was analyzed by BiG-SCAPE 2.0.0-beta.8 with '--record-type protocluster', '--classify class',

**Table 2 | Run steps extracted from log files used to define the major run tasks**

| Step | BiG-SCAPE 1.1.9 Start | BiG-SCAPE 1.1.9 Stop | BiG-SCAPE 2.0.0 Start | BiG-SCAPE 2.0.0 Stop | Notes |
|---|---|---|---|---|---|
| Input Parsing | The creation time of the "parameters.txt" file (the first file written during the analysis). | The modification time of the last modified fasta cache file. | The timestamp in a log containing the text "Starting BiG-SCAPE". | The timestamp in a log containing the text "First task: TASK.HMM_SCAN". | This step includes the reading and parsing of input GBK files provided. |
| hmmscan | The modification time of the last modified fasta cache file. | The modification time of the last modified domtable output file from hmmscan. | The timestamp in a log containing the text "First task: TASK.HMM_SCAN". | The timestamp in a log containing the text "DB: HSP save done at". | For the two HMMER-related steps, BiG-SCAPE 1.1.9 executes hmmscan and hmmalign as subprocesses using their respective binary files on the system. |
| hmmalign | The modification time of the last modified domtable output file from hmmscan. | The modification time of the last.pfd file was taken as the runtime. | The timestamp in a log containing the text "DB: HSP save done at". | The timestamp in a log containing the text "DB: HSP alignment save done at". | For the two HMMER-related steps, BiG-SCAPE 2.0.0 performs hmmscan and hmmalign by using the PyHMMER Python package [49]. |
| Distance generation | The modification time of the last.pfd file. | The modification time of the mix network.network file containing all edges. | Timestamp in a log containing the text "DB: HSP alignment save done at". | The timestamp in a log containing the text "Generating families". | This includes the computational time spent on the all-vs-all distance calculation. |
| GCF calling | The modification time of the mix network.network file. | The modification time of the "runtimes.txt" log file (the last file written during the analysis). | The timestamp in a log containing the text "Generating families". | The log timestamp in a log containing the text "All tasks done at". Log Timestamp containing "Generating GCF alignments" comprises all time spent prior to output file generation. | In BiG-SCAPE 1.1.9, this includes connected component generation, GCF calling, and any output file generation still left to be carried out. To enable comparison, output file generation was appended to the timestamp for BiG-SCAPE 2.0.0. |

Times noted for BiG-SCAPE 2.0.0 include time spent saving results to the database.

'--profiling' and 'CONSERVE_MEMORY=True' (conserves memory at the cost of speed), using 128 cores on a 3TB server, and with BiG-SLiCE 2.0.0 with default parameters using 64 cores on a 3TB server.

The number of input genomes and BGC regions were determined by counting, respectively, the number of downloaded genome folders and processed antiSMASH GBK files. The total runtime, number of computed connected components and number of bins were determined from the BiG-SCAPE log file. Similarly, BiG-SLiCE runtime was determined from the log in the output database by querying "select * from run_log". The average and peak RAM was obtained through the monitoring software Ganglia[59]. The number of GCFs, excluding singletons, was obtained by querying the BiG-SCAPE output database with "select count(*) from family", and the BiG-SLiCE output database with "select count(distinct gcf_id) from gcf_membership where gcf_id in (select gcf_id from gcf_membership group by gcf_id having count(*) >1)". The number of unique classes could be found in the antiSMASH database statistics page.

IGUA v0.1 was run on a concatenated file of all Antismash DB v4.0b2 cluster GBKs wherein locus IDs were altered to be unique across all clusters. This run was executed using 64 cores on a 3TB server. Because IGUA reports no runtimes of its own in its logging, the runtimes for IGUA were measured by using the standard unix "time" utility.

### Benchmarking biological accuracy

To evaluate and compare biological accuracy between different tools and tool versions, we obtained eight publicly available datasets with manually curated family assignments resulting from published results (Supplementary File 1)[13,18,60–65]. Included in these is the curated GCF assignments of MIBiG version 1.3 used to benchmark both BiG-SCAPE 1.0 and BiG-SLiCE 1.0.[18,19] Furthermore, we generated curated family assignments for a ninth dataset (dataset B (Divergent), see Supplementary File 2) by selecting a number of publicly available BGCs with high level of similarities in their gene architectures as well as produced metabolite structures, but encoded in the genomes of distantly related bacterial taxa. This dataset also features singletons from the featured taxa and compound classes (Supplementary File 2). Overall, these datasets consist of GenBank files generated by different versions of antiSMASH and vary wildly in size and composition (Supplementary File 1).

All nine curated datasets were analyzed by BiG-SLiCE version 1.1.1 and 2.0.0 with default parameters while increasing thresholds ranging from 0 to 1500 and 0 to 1.2, respectively. BiG-SCAPE version 1.1.9 was ran with both '--mode global' and '--mode glocal' with additional parameters '--cutoffs 0.1 0.2 0.3 0.4 0.5 0.6 0.7 0.8 0.9 1 --include_singletons --mix --no_classify --include_gbk "*"'. BiG-SCAPE 2.0.0-beta.1 was ran with all combinations of '--alignment-mode' (global, glocal, local) and '--extend-strategy' (legacy, simple match, greedy) with additional parameters '-gcf-cutoffs 0,0.1,0.2,0.3,0.4,0.5,0.6,0.7,0.8,0.9,1 --include-singletons -mix -classify none -include-gbk "*" -no-trees'.

To run IGUA v0.1, all GenBank files belonging to each of the nine benchmark datasets were concatenated into one GenBank file, and LOCUS names were changed to their original filename, excluding file extension. IGUA was subsequently run repeatedly on each file with default settings while increasing '--clustering-distance' from 0 to 1.0 in 0.1 increments. Output tsv files were modified to only keep the "cluster_id" and "gcf_id" columns for compatibility with downstream processing.

All output was subsequently analyzed by the BiG-SCAPE 2.0.0 Benchmark workflow, which compares the curated GCF assignments to the computed GCF assignments. Here, V-measure was compared at the cutoff where the number of computed GCFs best matched the number of curated GCFs to ensure a valid comparison between different tools and clusterings.

**Reporting summary.** Further information on research design is available in the Nature Portfolio Reporting Summary linked to this article.

## Data availability

The benchmarking datasets, curated assignments, BiG-SCAPE and BiG-SLiCE clustering output, and BiG-SCAPE benchmark output have been deposited in the Zenodo database and are accessible via https://zenodo.org/records/16838819. BiG-SCAPE 2.0 clustering output for MIBiG BGCs used to generate Supplementary Fig. S4 and Supplementary Table S1 have been deposited in the Zenodo database and are accessible via https://zenodo.org/records/16838819. BiG-SCAPE 2.0 Cluster and BiG-SLiCE 2.0 output for the antiSMASH database 4.0 have been deposited on Zenodo and are accessible via https://zenodo.org/records/16838819. Source data are provided in this paper.

## Code availability

The code for BiG-SCAPE is publicly available and has been deposited in the BiG-SCAPE GitHub repository[66] at https://github.com/medema-group/BiG-SCAPE under GNU Affero General Public License v3.0. Version 2.0.0-beta.1 associated with this publication is archived in Zenodo and is accessible via https://zenodo.org/records/17865734[67]. Version 2.0.0-beta.8 associated with this publication is archived in Zenodo and is accessible via https://zenodo.org/records/17865391[68]. The code for BiG-SLiCE is publicly available and has been deposited in the BiG-SLiCE GitHub repository at https://github.com/medema-group/bigslice under GNU Affero General Public License v3.0. The specific version of the code associated with this publication is archived in Zenodo and is accessible via https://zenodo.org/records/10783493[69]. Supplementary scripts used to generate Figs. 2–3, Supplementary Tables S3, S4, Supplementary Figs. S8–S10, and Supplementary Data file 3 can be accessed from Zenodo (https://zenodo.org/records/16838820).

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

## Acknowledgements

We thank M. Adamek, A. Gavriilidou, and N. Ziemert (*Amycolatopsis*, Reclassification) as well as M.M. Zdouc (*Planomonospora*) for their help in providing and compiling curated benchmarking datasets. The BiG-SLiCE developers thank Matin Nuhamunada, Amay Agrawal, and Friederike Biermann for their contributions of bug fixes to the V2 codebase. This work was supported by the open science framework of the Dutch Research Council (NWO), project 'BiG-CODEC' - OSF.23.1.044, an ERC Starting Grant 948770-DECIPHER to M.H.M, and the Novo Nordisk Foundation AEGIS (NNF24SA0092560) to N.L.L.L. The U.S. Department of Energy Joint Genome Institute (https://ror.org/04xm1d337), a DOE Office of Science User Facility, is supported by the Office of Science of the U.S. Department of Energy [DE-AC02-05CH11231].

## Author contributions

A.D., C.L., and N.L.L.L.: Conceived and designed BiG-SCAPE 2.0 algorithms, developed the BiG-SCAPE 2.0 software, designed and performed the benchmarks and data analysis, wrote BiG-SCAPE 2.0 documentation, drafted and wrote the manuscript. S.A.K.: Conceived and designed BiG-SLiCE-2.0-related algorithms and developed the BiG-SLiCE 2.0 software, drafted and wrote the manuscript. J.C.N.-M.: Guided interpretation of BiG-SCAPE 1 code, gave feedback on directions and improvements, compiled and provided curated benchmarking datasets (JK1, Metcalf). D.T.D: Developed the BiG-SLiCE 2.0 software. N.J.M.: Program management, resourcing. M.H.M: Project management, funding acquisition, guided the design of algorithms, benchmarks and analysis, compiled and provided curated benchmarking datasets (Divergent, *Fusarium*, *Rhodococcus*). All authors edited the manuscript before submission.

## Competing interests

M.H.M. is a member of the Scientific Advisory Boards of Hexagon Bio and Hothouse Therapeutics Ltd. The remaining authors declare no competing interests.
