## [Transparent Peer Review file · Nature Communications]

BiG-SCAPE 2.0 and BiG-SLiCE 2.0: scalable, accurate and interactive sequence clustering of metabolic gene clusters

Corresponding Author: Professor Marnix Medema

Version 0:

Reviewer comments:

Reviewer #1

(Remarks to the Author)

Title: BiG-SCAPE 2.0 and BiG-SLiCE 2.0: scalable, accurate and interactive sequence clustering of metabolic gene clusters

Manuscript #: NCOMMS-25-65797-T

Recommendation: Major revision

This manuscript presents BiG-SCAPE 2.0 and BiG-SLiCE 2.0, two major updates to widely used software for biosynthetic gene cluster (BGC) clustering and classification. The improvements in algorithmic performance, scalability, and usability are clear and provide substantial value to the genome mining community. Importantly, the authors demonstrate large-scale applicability on the entire antiSMASH-DB, highlighting the relevance of the tools for global natural product discovery efforts.

The manuscript is strong in its technical contributions, but certain areas require clarification, additional benchmarking, and a more balanced interpretation of results. The manuscript presents an important update to community-standard tools, but clarifications and stronger contextualization are necessary before publication. I recommend major revision before acceptance. Here are the major and minor comments below:

Major comments:

1. The new alignment modes greatly increase flexibility, but the manuscript does not clearly indicate when users should prefer each mode or how these choices influence outcomes. A more systematic evaluation or practical guidelines would be valuable for reproducibility.
2. In Figure 2, certain datasets (especially, Dataset B and H) show a decrease in V-measure with the updated version. In my opinion, the author does not provide sufficient explanation for why these cases behave differently. A deeper discussion of these exceptions—whether due to dataset heterogeneity, limitations of cosine similarity, or characteristics of the curation—would be important to help users understand the contexts in which BiG-SLiCE 2.0 may underperform.
2-1) Plus, the statement ‘average V-measure increase of 17%, especially ‘datasets particularly associated with shorter BGCs such as RiPPs (106% increase)’ could be misleading. As written, it might suggest that RiPPs have lower improvement, whereas in fact the intended meaning is that RiPPs exhibit a much larger relative gain compared to the average. I recommend clarifying whether these percentages refer to absolute or relative improvements, and rephrasing to avoid misinterpretation.
2-2) The manuscript should more carefully contextualize the reported gains. A 17% improvement for RiPPs is indeed substantial, but the average global increase of ~6% is more modest. The narrative should reflect this distinction to avoid overstating the impact.
3. Benchmarking is mainly against earlier versions of BiG-SCAPE and BiG-SLiCE. While understandable, it would be valuable to discuss how these tools compare with recent alternatives (e.g., DeepBGC, GECCO, NeuRiPP). If direct comparisons are not feasible, acknowledging relative scope and complementarity would improve positioning.
4. The authors convincingly demonstrate scalability on antiSMASH-DB (~260,000 regions). Given that the same group has also curated very large-scale datasets (e.g., BiG-FAM, comprising millions of BGCs), it would be highly valuable to discuss how BiG-SCAPE/SLiCE 2.0 would perform at such metagenomic scale. Even if new benchmarks are not feasible within this manuscript, a discussion of expected limitations and computational requirements for million-scale datasets would strengthen the work.
5. As this is primarily a methodological work, experimental validation is not strictly necessary. However, if it is possible, the manuscript would be strengthened by highlighting literature-based validations—e.g., showing that the clustering of well-

characterized BGC families aligns with experimentally known metabolite families. This would help readers appreciate the biological relevance of the improved clustering without requiring new wet-lab data.

6. The manuscript presents BiG-SCAPE 2.0 and BiG-SLiCE 2.0 as powerful clustering and classification tools. While the focus here is methodological, these improvements could potentially facilitate downstream compound discovery workflows—for instance, by prioritizing novel BGC families for experimental validation or integrating with predictive frameworks such as DeepBGC.

- Do the authors envision future directions where BiG-SCAPE/SLiCE could be directly applied or extended toward compound discovery pipelines?

- Are there plans to integrate these clustering results with databases of known metabolites (e.g., MIBiG) to systematically identify unexplored chemical diversity?

7. As BiG-SCAPE and BiG-SLiCE are expected widely used community standards, their long-term sustainability is a critical issue. The manuscript currently does not address how future updates (such as antiSMASH upgrades, and database expansion) will be managed. It would strengthen the paper if the authors could:

7-1) Clarify whether there is an active plan to keep the tools synchronized with future Pfam/antiSMASH updates.

7-2) Indicate if there is a structured mechanism for community contributions (e.g., GitHub issue tracking, pull requests, or open governance).

7-3) Discuss how long-term maintenance will be ensured—whether through dedicated funding, integration with larger consortia, or continued lab-based support.

Providing these details would reassure users that BiG-SCAPE/SLiCE will remain reliable and up to date, which is particularly important given their status as community-standard resources.

Minor Comments

- Clearer documentation and usage guidelines (especially on parameter choices) would improve usability for non-expert users.

(Remarks on code availability)

Reviewer #2

(Remarks to the Author)

Dear Authors,

BiG-SCAPE 2.0 and BiG-SLiCE 2.0 were developed for natural product research. These programs provide more accurate and less time-consuming analysis than previous programs, and will continue to assist researchers in studying the diversity and evolution of specialized microbial metabolisms, as well as in efforts to discover new biological compounds.

I have a few comments.

Line 412, *Rhodococcus* is Genus, it should be italic letter.

In Supplementary Table S4, Scientific name should be correct. For example, *Streptomyces* (Genus, should be italic letter) sp. (should be not italic letter).

CBMAI 2042 (strain should be not italic letter). Please edit all carefully.

Regards,
Reviewer

(Remarks on code availability)

Reviewer #3

(Remarks to the Author)

Genome mining of biosynthetic gene clusters (BGCs) has become an important area in natural product (NP) discovery and drug development. Clustering BGCs into gene cluster families provides valuable insights into the chemical diversity of NPs.

With the rapid expansion of genomic datasets and the introduction of new concepts and standards related to BGCs, there is a growing demand for clustering tools that ensure accuracy, speed, and scalability. In this work, the authors present updated versions of two previously developed BGC clustering tools: BiG-SCAPE, designed for interactive sequence similarity network analysis of BGCs, and BiG-SLiCE, developed for clustering large numbers of BGCs. These updates represent meaningful advances that enhance the applicability of both tools in current BGC research. Thanks to the author's efforts.

Two points the authors may consider:

1. Could a strategy be developed to integrate the advantages of the two tools, rather than maintaining them as separate,

independent resources? In practice, it can be difficult for users to decide which tool is more suitable for a particular task, especially given the differences in accuracy between the two approaches.

2. In the final section of the manuscript, the authors generated two classifications of 260,630 biosynthetic regions from the antiSMASH database using the updated tools. While this demonstrates the functionality of the methods, I think that to establish a high-precision, consensual classification for query and visualization would likely represent an even more impactful contribution. I know the current work focuses on updating clustering tools, I look forward to seeing the authors further develop this direction in future studies.

(Remarks on code availability)

Both codes are well-established and are accompanied by clear and informative instructions.

The installation process for BiG-SCAPE is straightforward, and all four workflows run smoothly with the tutorial dataset. However, when using the query workflow, the query node does not appear to be highlighted in the network.

During the installation of BiG-SLiCE, I encountered an error related to generating package metadata, which prevented further testing of the code. This issue may be associated with configuration settings on my system.

Version 1:

Reviewer comments:

Reviewer #1

(Remarks to the Author)

The authors have satisfactorily addressed all major and minor comments raised in the previous review. The revised manuscript provides clear explanations and appropriate revisions in response to the earlier concerns, including improved benchmarking details, clarification of performance differences in specific datasets, and a more balanced interpretation of accuracy metrics.

I therefore consider that the authors have adequately addressed the reviewers' points, and I recommend that the manuscript be accepted for publication in Nature Communications after editorial assessment.

(Remarks on code availability)

Reviewer #3

(Remarks to the Author)

My comments have been fully answered.

(Remarks on code availability)

BiG-SCAPE 2.0 & BiG-SLiCE 2.0 Manuscript - Rebuttal

Rebuttal Comments Reviewer 1:

[...]“The new alignment modes greatly increase flexibility, but the manuscript does not clearly indicate when users should prefer each mode or how these choices influence outcomes. A more systematic evaluation or practical guidelines would be valuable for reproducibility.”[...]

The authors thank the reviewer for the comment, and agree that while the new alignment modes increase flexibility and accuracy, that the increased number of running parameter options creates new challenges with regards to case by case parameter optimisation and reproducibility of results. However, a systematic evaluation of the performance and accuracy of all combinations of run parameters, applied to an extensive set of curated datasets that comprehensively span BGC classes and other characteristics, is unfortunately infeasible given its combinatorial complexity. As the results show, the authors are confident that both BiG-SCAPE 2.0 and BiG-SLiCE 2.0, at their default run parameters, deliver accurate clusterings that hold biological meaning. As such, our advice is that users start by running the tools at their default parameters, and if resource availability allows it, to go beyond the default run and explore how the newly added flexibility can support their analysis. Additionally, the relevant section where the alignment modes are described and practical guidelines are given for their usage has been updated, in page 9 of the manuscript as follows: “[...] Alternatively, the ‘global’ mode where all domains of each gene cluster record are compared may be preferred for datasets of gene clusters with no contig breaks, and manually curated cluster borders. As the ‘global’ mode finds a compromise between the ‘local’ and ‘global’ modes, it is used by BiG-SCAPE 2 as the default mode. [...]”

[...]“In Figure 2, certain datasets (especially, Dataset B and H) show a decrease in V-measure with the updated version. In my opinion, the author does not provide sufficient explanation for why these cases behave differently. A deeper discussion of these exceptions—whether due to dataset heterogeneity, limitations of cosine similarity, or characteristics of the curation—would be important to help users understand the contexts in which BiG-SLiCE 2.0 may underperform.”[...]

One drawback of the linear clustering method used in BiG-SLiCE is that it is less deterministic than an all-to-all approach used in BiG-SCAPE. This is similar to the CD-HIT approach in protein clustering. Depending on the nature of the data and the input order in the algorithm, the actual GCF results may change. Dataset H represents a set of BGCs with (subsets of) mutually highly similar gene compositions and architectures, with only minor differences. Dataset B

represents the opposite scenario, where the sequences are so divergent that very few domains are shared at the subPfam level. Although we deliberately did not want to add speculative discussions in the manuscript on why datasets B and H showed BiG-SLiCE v2 performing worse than v1, we would hypothesise that BiG-SLiCE v2's cosine distance underperforms at both extremes — when vectors share too few features or when they share too many (with only small differences). In the former case, sparse or mostly non-overlapping feature vectors yield artificially low cosine similarity despite underlying structural relatedness, while in the latter, near-parallel vectors produce little angular separation, obscuring subtle but biologically meaningful differences.

[...]“Plus, the statement ‘average V-measure increase of 17%, especially ‘datasets particularly associated with shorter BGCs such as RiPPs (106% increase)’ could be misleading. As written, it might suggest that RiPPs have lower improvement, whereas in fact the intended meaning is that RiPPs exhibit a much larger relative gain compared to the average. I recommend clarifying whether these percentages refer to absolute or relative improvements, and rephrasing to avoid misinterpretation.”[...]

The authors thank the reviewer for the comment and have updated this passage to clarify the type of percentage and what they reference on page 13 of the manuscript as follows: [...]“BiG-SLiCE 2.0 showcases improved accuracy when compared to its previous version (average relative V-measure increase of 17%), with the largest increases seen especially in datasets particularly associated with shorter BGCs such as RiPPs (Dataset G: 106% relative V-measure increase; Fig 2.)”[...].

[...]“The manuscript should more carefully contextualize the reported gains. A 17% improvement for RiPPs is indeed substantial, but the average global increase of ~6% is more modest. The narrative should reflect this distinction to avoid overstating the impact.”[...]

The authors thank the reviewer for the comment and recognize that, in line with the previous comment, this section was unclear, and have updated this passage (on page 13, see comment above) to better reflect the 17% average relative global increase and 106% relative increase for RiPPs.

[...]“Benchmarking is mainly against earlier versions of BiG-SCAPE and BiG-SLiCE. While understandable, it would be valuable to discuss how these tools compare with recent alternatives (e.g., DeepBGC, GECCO, NeuRiPP). If direct comparisons are not feasible, acknowledging relative scope and complementarity would improve positioning.”[...]

The authors thank the reviewer for the comment and agree with the value in benchmarking against the most recent alternatives. Complementarity with BGC detection tools like DeepBGC

and GECCO is assured via antiSMASH's side-loading functionality, as well as BiG-SCAPE 2's --force-gbk parameter which allows non antiSMASH processed GBK files to be used as input. We should emphasise that DeepBGC, GECCO and NeuRiPP do not perform the same task as BiG-SCAPE and BiG-SLiCE, i.e. clustering BGCs into gene cluster families. Hence, a direct comparison is indeed not possible.

Yet, as a further courtesy to the reviewer and to the scientific community, we now compared our tools to the only comparable tool (which has been developed in parallel and has recently become available as a preprint), IGUA (<https://doi.org/10.1101/2025.05.15.654203>). The manuscript section has been updated on pages 18 and 19 (“[...] Finally, while all previously described benchmarking has been carried out against earlier versions of the tools, we additionally leverage the antiSMASH database to compare our tools with the most recent alternative gene cluster clustering tool, IGUA [...] Future work could explore potentially merging MMseqs2-based features with vector clustering and/or sequence similarity networking to combine the advantages of each method. [...]”), see also Supplementary Fig. S8 and S10.

IGUA is mostly similar to BiG-SLiCE instead of BiG-SCAPE, as it also aims to provide direct clustering of input BGCs without interactive sequence similarity network visualizations. It gains considerable speed by replacing the use of pHMMs by MMseqs2 clustering of protein sequences, although theory suggests that its use of an all-vs-all distance matrix and hierarchical clustering will scale less favorably with very large datasets that exceed the antiSMASH database in size. Furthermore, such choice of algorithm does not allow direct assignment of new BGCs to pre-processed GCFs, which is a very useful feature of the vector clustering approach used in BiG-SLiCE, demonstrated by the wide usage of our BiG-FAM database.

So altogether, IGUA provides a useful complementary algorithm with very fast clustering of medium-to-large datasets, even though BiG-SLiCE v2 and BiG-SCAPE v2 still provide unique capabilities with larger theoretical scalability and the ability to place BGCs into precomputed GCFs (for BiG-SLiCE) and interactive network visualization and exploration (for BiG-SCAPE). Future work could explore potentially merging MMseqs2-based features with vector clustering and/or sequence similarity networking to combine the advantages of each method.

[...]“The authors convincingly demonstrate scalability on antiSMASH-DB (~260,000 regions). Given that the same group has also curated very large-scale datasets (e.g., BiG-FAM, comprising millions of BGCs), it would be highly valuable to discuss how BiG-SCAPE/SLiCE 2.0 would perform at such metagenomic scale. Even if new benchmarks are not feasible within this manuscript, a discussion of expected limitations and computational requirements for million-scale datasets would strengthen the work.”[...]

The authors thank the reviewer for the comment. Indeed, we agree that running BiG-SLiCE 2.0 at the BiG-FAM scale would be interesting, and we are planning such an analysis for the future, i.e., for the eventual update of the database. Due to the updates mentioned in the manuscript, we saw that BiG-SLiCE 2.0 was already 1.5x faster than the original, and more speed can be

gained as the database grows larger (due to the I/O bottleneck in v1). Therefore, we expect that BiG-SLiCE 2.0 will perform comparably well for such a couple million BGCs sized dataset. However, while BiG-SLiCE (both versions) will comfortably handle up to several million BGCs with a 128 GB RAM server, handling datasets with sizes beyond ten million BGCs, i.e., the 13 million BGCs from the Secondary Metabolite Collaboratory (SMC) database, will probably require more specialized hardware.

For the sake of conciseness, we addressed this in the manuscript, page 18, by adding the following sentence in the discussion of the antiSMASH database runtimes: “Based on the resource use, this dataset size likely approaches the maximum for BiG-SCAPE on most standard server hardware architectures, whereas BiG-SLiCE remains a viable choice for datasets scaling into millions of BGCs. (...)”

[...]“As this is primarily a methodological work, experimental validation is not strictly necessary. However, if it is possible, the manuscript would be strengthened by highlighting literature-based validations—e.g., showing that the clustering of well-characterized BGC families aligns with experimentally known metabolite families. This would help readers appreciate the biological relevance of the improved clustering without requiring new wet-lab data.”[...]

The authors thank the reviewer for the comment and acknowledge the value of experimentally validated datasets, which motivated the inclusion of Dataset F (Metcalf), which is based on experimentally validated metabolite similarity. The flow of discussions related to the curated GCF assignment benchmarks, in particular with regards to Dataset F, has been improved to better highlight how BGC clustering by BiG-SCAPE 2.0 and BiG-SLiCE 2.0 is in agreement with experimentally verified metabolite/BGC families, see page 15 “[...] Finally, Dataset F (Metcalf) features curated GCF assignments based on the molecular similarity of experimentally characterized metabolites produced by their BGCs (Supplementary Table S3), and showcases how BiG-SCAPE 2.0 is consistently able to connect the genomic biosynthetic landscape to the chemical space of produced metabolites across run parameters. [...]”

[...]“The manuscript presents BiG-SCAPE 2.0 and BiG-SLiCE 2.0 as powerful clustering and classification tools. While the focus here is methodological, these improvements could potentially facilitate downstream compound discovery workflows—for instance, by prioritizing novel BGC families for experimental validation or integrating with predictive frameworks such as DeepBGC.

**- Do the authors envision future directions where BiG-SCAPE/SLiCE could be directly applied or extended toward compound discovery pipelines?
- Are there plans to integrate these clustering results with databases of known metabolites (e.g., MIBiG) to systematically identify unexplored chemical diversity?”[...]**

The authors thank the reviewer for the comment. Indeed, BiG-SCAPE and BiG-SLiCE are used in the wider compound discovery workflows, facilitating prioritization of novel GCFs, GCFs

related to known compounds (as those in the MIBiG database), as well as pairing of multimodal data such functional gene annotation data (e.g. resistance genes), environmental metadata, metabolome profiles and bioactivity data. BiG-SCAPE is already integrated in such pipelines, as is the cases of NPlinker and BGCFLOW. As suggested, we also envision future directions where BiG-SCAPE and BiG-SLiCE are further integrated in gene cluster predictive frameworks and compound discovery pipelines, along with other paired data modalities such as bioactivity data. To address the second point, we agree with the reviewer that there is tremendous value to integrating clustering results in a global map of diversity that includes the experimentally validated MIBiG BGCs as well as predicted gene clusters from databases such as antiSMASH DB, and beyond. In the coming years, we hope to construct such a resource and make it publicly available to the research community.

[...]“As BiG-SCAPE and BiG-SLiCE are expected widely used community standards, their long-term sustainability is a critical issue. The manuscript currently does not address how future updates (such as antiSMASH upgrades, and database expansion) will be managed. It would strengthen the paper if the authors could: Clarify whether there is an active plan to keep the tools synchronized with future Pfam/antiSMASH updates.”[...]

The authors thank the reviewer for the comment, and agree that long term sustainability is an important issue. Synchronizing both tools with future updates of the commonly used gene cluster detection tools has been made easier by flexibly parsing input data, e.g. by moving away from the previous hard-coded antiSMASH classification and facilitating use of non SMASH-tools processed gene clusters. Additionally, the tools can use any version of the Pfam database, as well as alternative (Pfam-style formatted) collections of pHMMs for protein domain detection. We have expanded the software sustainability section on page 11 of the manuscript to address these points “[...] Furthermore, with regard to software sustainability [...] community contributions as well as user feedback, which is welcomed and can be done easily through the Github page. [...]”.

[...]“Indicate if there is a structured mechanism for community contributions (e.g., GitHub issue tracking, pull requests, or open governance).”[...]

The authors thank the reviewer for the comment, and indeed this structured mechanism is in place using Github, where both tools are hosted. We have expanded the software sustainability section on page 11 of the manuscript to address this point (see comment above).

[...]“Discuss how long-term maintenance will be ensured—whether through dedicated funding, integration with larger consortia, or continued lab-based support. Providing these details would reassure users that BiG-SCAPE/SLiCE will remain reliable and up to date, which is particularly important given their status as community-standard resources.”[...]

The authors thank the reviewer for the comment and agree that long-term maintenance is of high priority. This task will fall on the permanent Research Software Engineer positioned in the Bioinformatics Group at Wageningen University. The software sustainability section on page 11 of the manuscript has been expanded with respect to continued development (see comment above).

[...]“Clearer documentation and usage guidelines (especially on parameter choices) would improve usability for non-expert users.”[...]

The authors thank the reviewer for the comment and have updated the software sustainability section on page 11 of the manuscript with reference to the wiki attached to the BiG-SCAPE github page (see comment above).

Rebuttal Comments Reviewer 2:

[...]“Line 412, Rhodococcus is Genus, it should be italic letter.”[...]

The authors thank the reviewer for the comment and have updated this on page 14 of the manuscript.

[...]“In Supplementary Table S4, Scientific name should be correct. For example, Streptomyces (Genus, should be italic letter) sp.(should be not italic letter).”[...]

The authors thank the reviewer for the comment and have updated Supplementary Table 4 accordingly on page 7 of the supplemental tables.

[...]“CBMAI 2042 (strain should be not italic letter). Please edit all carefully.”[...]

The authors thank the reviewer for the comment and have updated Supplementary Table 4 accordingly on page 7 of the supplemental tables.

Rebuttal Comments Reviewer 3:

[...]“Could a strategy be developed to integrate the advantages of the two tools, rather than maintaining them as separate, independent resources? In practice, it can be difficult

for users to decide which tool is more suitable for a particular task, especially given the differences in accuracy between the two approaches.”[...]

The authors thank the reviewer for the comment, and indeed, we agree that this scenario would be ideal. While this would require a large new software engineering project to turn it into reality, we hope to be able to address this need in the future. At the moment, we advise users to choose either tool based on their requirements with regards to input dataset size, and available computational resources, i.e. to prefer BiG-SCAPE for small to mid-sized datasets, and BiG-SLiCE for large datasets. We address the choice between tools in the antiSMASH-DB section on page 17 of the manuscript as follows: [...]”Based on the resource use, this dataset size likely approaches the maximum for BiG-SCAPE, where BiG-SLiCE remains as a viable choice for million-scale datasets.”[...] (see also response to reviewer #1).

[...]”In the final section of the manuscript, the authors generated two classifications of 260,630 biosynthetic regions from the antiSMASH database using the updated tools. While this demonstrates the functionality of the methods, I think that to establish a high-precision, consensual classification for query and visualization would likely represent an even more impactful contribution. I know the current work focuses on updating clustering tools, I look forward to seeing the authors further develop this direction in future studies.”[...]

The authors thank the reviewer for the comment, and agree that it would be valuable to generate a single consensual map of global diversity that can be effectively queried and interacted with. Although, as the reviewer accurately points out, this is beyond the scope of this publication, there is an ongoing project (<https://research-software-directory.org/projects/big-views>) at Prof. Medema’s group that aims to generate this resource in the future and make it publicly available to the research community.

[...]”The installation process for BiG-SCAPE is straightforward, and all four workflows run smoothly with the tutorial dataset. However, when using the query workflow, the query node does not appear to be highlighted in the network.”[...]

The authors thank the reviewer for the comment, and have resolved the issue in the latest release of BiG-SCAPE (<https://github.com/medema-group/BiG-SCAPE/releases/tag/v2.0.0>).

[...]”During the installation of BiG-SLiCE, I encountered an error related to generating package metadata, which prevented further testing of the code. This issue may be associated with configuration settings on my system.”[...]

This error might have been caused by a failure in the pip installation environment, and not from the source code itself. We invite the reviewer to submit an example screenshot of the error report via the issues tracker (<https://github.com/medema-group/bigslice/issues>), and we will try to help them through.